# A non-Newtonian fluid quasi-solid electrolyte designed for long life and high safety Li-O$_2$ batteries

Guangli Zheng[1], Tong Yan[1], Yifeng Hong[2], Xiaona Zhang[1], Jianying Wu[2], Zhenxing Liang[1], Zhiming Cui [1], Li Du [1] & Huiyu Song [1] ✉

The Li dendrite growth and the liquid electrolyte volatilization under semi-open architecture are intrinsic issues for Li-O$_2$ battery. In this work, we propose a non-Newtonian fluid quasi-solid electrolyte (NNFQSE) SiO$_2$-SO$_3$Li/PVDF-HFP, which has both shear-thinning and shear-thickening properties. The component interactions among the sulfonated silica nanoparticles, liquid electrolyte, and polymer network are beneficial for decent Li$^+$ conductivity and high liquid electrolyte retention without volatilization. Furthermore, NNFQSE exhibits shear-thinning property to eliminate the stress of dendrite growth during repeated cycling. Meanwhile, when the force suddenly increases, such as a high current rate, the NNFQSE may dynamically turn shear-thickening to respond and mechanically stiffen to inhibit the lithium dendrite penetration. By coupling with the NNFQSE, the lithium symmetrical battery can run over 2000 h under 1 mA cm$^{-2}$ at room temperature, and the quasi-solid Li-O$_2$ battery actualizes long life above 5000 h at 100 mA g$^{-1}$.

In recent years, commercial Li-ion batteries are becoming more and more difficult to meet the needs of large-scale energy storage and long-lasting electric devices, even though they have close to the theoretical energy density (e.g., C/LiCoO$_2$ up to -387 Wh kg$^{-1}$)[1]. Li-O$_2$ batteries have aroused great research enthusiasm due to their ultra-high theoretical energy density of 3500 Wh kg$^{-1}$, nearly an order of magnitude higher than existing Li-ion batteries[2–4]. However, Li-O$_2$ batteries are still in their infancy and meet many obstacles that restrict their commercialization.

Owing to Li-O$_2$ batteries needing an open structure to allow oxygen in and out, there is the risk of liquid electrolyte leakage or volatilization and unsatisfactory cycling performance[5]. Up till now, a significant amount of effort has been devoted to the low volatile electrolyte[6], including but not limited to sulfamide-, sulfonamide-based electrolyte[7] and solvate ionic liquid [Li(G3)$_1$][TFSA], [Li(G3)$_4$][TFSA][8]. Yet low volatility does not mean non-volatile, the ionic conductivity and cycling life will eventually decrease with electrolyte consumption. In addition, the commonly used porous separator in the

traditional Li-O$_2$ battery, such as polypropylene (PP) or glass fiber (GF), can be punctured by uncontrollable lithium dendrites during long discharge/charge cycling, and then internal short-circuits and thermal runaway may be triggered[9–13]. Thus, it is urgent and crucial to develop a simple and effective approach to prevent the leakage and evaporation of liquid electrolyte and concurrently inhibit the dendrite growth and penetration for Li-O$_2$ batteries.

Surprisingly, some materials with non-Newtonian fluid properties are capable of balancing flexibility and protection against high-speed impacts, which can be used as shock absorber, cushioning system, and body armor[14,15]. It violates Newton's experimental law of viscosity since the relationship between shear stress and shear strain rate is not linear. Shear-thinning is an effect where a fluid's viscosity decreases with increasing shear stress, and the opposite is shear-thickening. Quanhong Yang et al.[16] have reported a quicksand-like semisolid state lithium metal anode with shear-thinning property by mixing lithium powder, carbon nanotubes, and poly(ethylene oxide) in electrolytes, which can eliminate the stress of dendrite growth and adapt the

[1]Guangdong Provincial Key Laboratory of Fuel Cell Technology, School of Chemistry and Chemical Engineering, South China University of Technology, Guangzhou 510641, China. [2]Department of Civil Engineering, South China University of Technology, Guangzhou 510641, China. ✉e-mail: hysong@scut.edu.cn

volume change of lithium metal in the plating/stripping process. Moreover, taking advantage of non-Newton rheological behavior, Yan's group[17] has used starch macromolecules to propose a "solid-liquid" interconvertible electrolyte for Zn metal battery. The non-Newtonian fluid electrolyte behaves fast electrochemical kinetics and well interface contact between electrode and electrolyte. Significantly, benefitted from its shear-thickened property, the electrolyte can mechanically harden to resist Zn dendrites in fast growing areas. Consequently, it is expected that the notion of non-Newtonian fluid will bring inspirations to multifunctional electrolyte.

In this work, we report a non-Newtonian fluid quasi-solid electrolyte (NNFQSE) with both shear-thinning and shear-thickening properties for Li-O₂ batteries. The shear-thinning property of NNFQSE can effectively absorb the stress caused by Li plating/stripping and cushion volumetric changes of Li anode. The shear-thickening property will rapid mechanically stiffen in response to instantaneous impact of macroscopic strain and inhibit the lithium dendrite growth. The NNFQSE is designed with component interaction between conventional polymer skeleton PVDF-HFP and modified Li⁺-conductive nanofillers SiO₂-SO₃Li (Fig. 1). Combined, honeycomb cross-linked network is formed through the ion-dipole interactions between the polar -CFx groups of PVDF-HFP and sulfo groups of SiO₂. The sulfonated silica nanoparticles (SiO₂-SO₃Li) with wrinkled radial structure can trap and retain plenty of liquid electrolytes inside, which can directly promote the transport of lithium ions, and prevent evaporation and leakage. What's more, the liquid electrolyte filled into composite polymer film can act as lubricant to allow the polymer molecules to slide against each other[17]. When the macroscopic strain increases or decreases during charging and discharging[18], the composite quasi-solid electrolyte exhibits non-Newtonian rheological behavior to protect the battery. As a result, the NNFQSE enables long-life performance for Li-O₂ cells above half a year (5000 h at 100 mA g⁻¹) without electrolyte leakage and lithium dendrites penetration. This study breathes life into the multifunctional electrolyte strategy for quasi-solid lithium-oxygen batteries and even other lithium batteries.

## Results and discussion
### Characterization of NNFQSE
The NNFQSE was prepared with PVDF-HFP and SiO₂-SO₃Li nanoparticles via tape casting process and described in the methods section in detail. Its digital photos are shown in Supplementary Fig. 1. It is worth noting that the ion-dipole interactions between the polar -CFx groups of PVDF-HFP and sulfo groups of SiO₂ make the NNFQSE

exhibit honeycomb morphology (Fig. 2a), which allows Li⁺ to travel much faster and accommodate more liquid electrolyte. For comparison, the pure PVDF-HFP displays disordered pores about 1 μm (Supplementary Fig. 2a, b), and the pores are almost blocked after adding unmodified SiO₂ (Supplementary Fig. 2c, d). Meanwhile, mercury intrusion porosimetry (MIP) measurements were carried out to characterize their porous features (Supplementary Table 1). PVDF-HFP is observed with the highest porosity ratio. After adding SiO₂ nanofillers into PVDF-HFP, the porosity ratio is decreased from 74.6% to 58.4%, and the volume density is increased from 0.34 g mL⁻¹ to 0.59 g mL⁻¹, which further prove the scanning electron microscopy (SEM) observations. For SiO₂-SO₃Li/PVDF-HFP, the porosity ratio, volume density, and total pore volume all fall in between PVDF-HFP and SiO₂/PVDF-HFP, revealing that the sulfonate fillers SiO₂-SO₃Li can make the skeleton matrix maintain certain porosity without being blocked. In addition, it is noticed that the total pore area increases from 7.56 m² g⁻¹, 16.06 m² g⁻¹ to 18.23 m² g⁻¹, respectively. This result benefits from the wrinkled radial structure and beauteous folded spherical morphology of SiO₂ nanofillers (Supplementary Fig. 3). And there also distribute a great deal of hierarchical pores on the particles. After modification, the SiO₂-SO₃Li fillers maintain the wrinkled radial structure (Fig. 2b). Furtherly, the surface channels of SiO₂-SO₃Li become more than twice deeper and richer (Fig. 2c) than SiO₂ (Supplementary Fig. 3c), which can explain the largest total pore area of SiO₂-SO₃Li/PVDF-HFP. These changes also made a prodigious contribution to increase the contact area and trap more liquid electrolyte inside. And the elements Si, O of SiO₂ and Si, O, S of SiO₂-SO₃Li are uniformly distributed which can be clearly seen in the EDS elements mapping (Supplementary Figs. 3–5). The S signal indicates that -SO₃Li groups have successfully loaded on SiO₂ hydroxyl chain. We also measured the content of S (6184 mg kg⁻¹) and Li (862 mg kg⁻¹) in the sample through Inductively Coupled Plasma Mass Spectrometry (ICP-MS, Thermo scientific iCAP 7200 Duo) that further confirmed the existence of -SO₃Li groups.

Fourier Transform Infrared Spectroscopy (FTIR) was conducted to investigate surface chemistry. As shown in Supplementary Fig. 6, the region of 1000-1300 cm⁻¹ corresponding to Si−O−Si stretching adsorption turns widen while in SiO₂-SO₃H and SiO₂-SO₃Li, and the characteristic peak of -SO₃⁻ stretching at 1190 cm⁻¹ indicates the sulfonated treatment[19,20]. It's clearly shown in Fig. 2d that the -SO₃⁻ stretching at 1190 cm⁻¹ becomes more obvious and Si−O−Si antisymmetric stretching vibration of SiO₂-SO₃Li experiences a 5 cm⁻¹ redshift (1095 cm⁻¹ to 1100 cm⁻¹) after combining with PVDF-HFP, verifying the ion-dipole interactions between them. Moreover, the S 2p signal in XPS

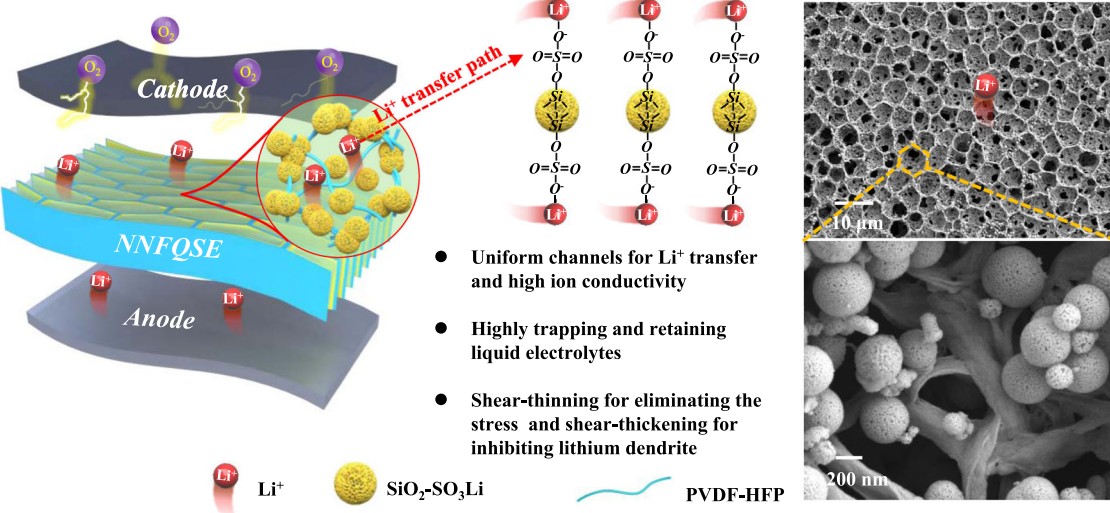

**Fig. 1** | Schematic diagram of NNFQSE SiO₂-SO₃Li/PVDF-HFP with uniform Li⁺ transport channels.

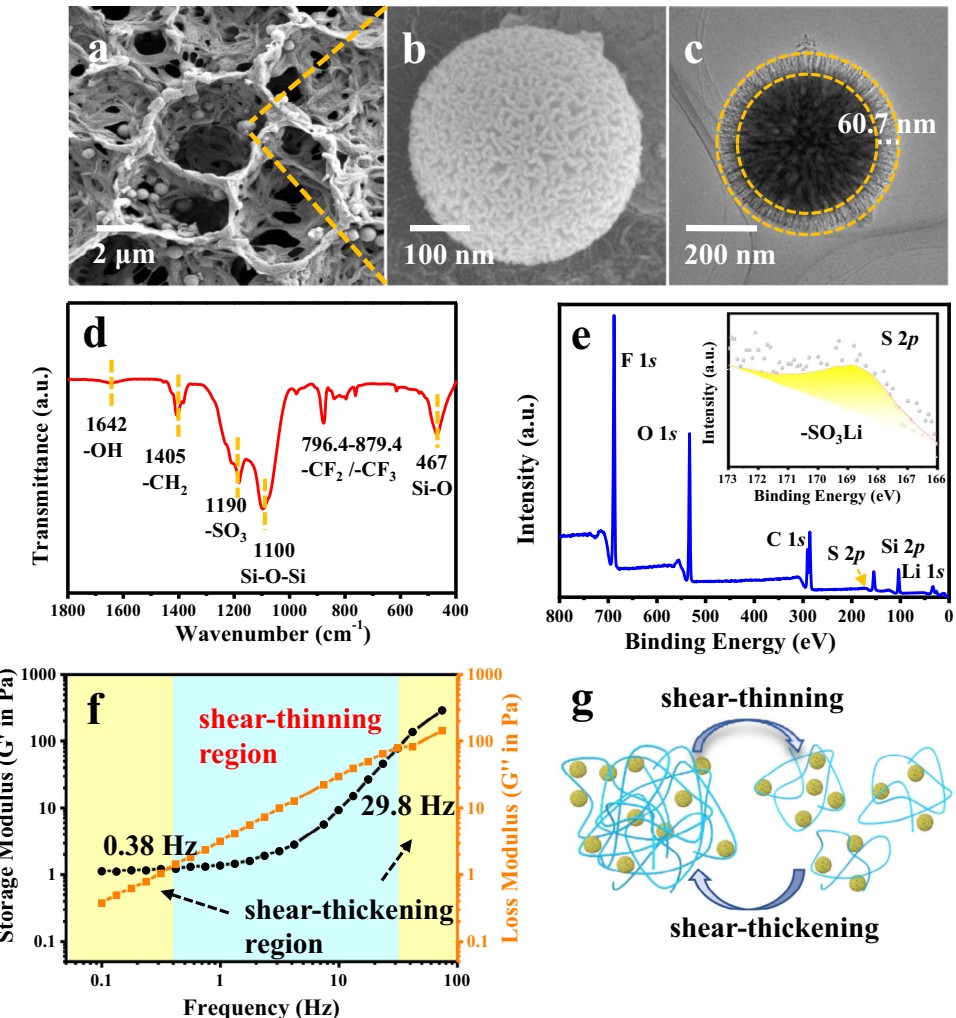

**Fig. 2 | Characterization of SiO₂-SO₃Li/PVDF-HFP quasi-solid electrolyte.**
**a**, **b** SEM analysis; **c** TEM analysis of the SiO₂-SO₃Li particles; **d** FTIR spectra; **e** XPS full spectrum and S 2*p* spectra; **f** Rheological analysis of NNFQSE; **g** Shear-responsive process, the blue lines represent PVDF-HFP and the yellow particles represent SiO₂-SO₃Li.

full spectrum and the -SO₃Li peak in S 2*p* spectra provide evidence that the -SO₃Li groups have been grafted on the surface of SiO₂ successfully (Fig. 2e). In addition, XRD was used to further verify how fillers effect on PVDF-HFP (Supplementary Fig. 7). PVDF-HFP exhibits four peaks at 18.5°, 20.2°, 26.6°, and 38.8°, which are assigned to its typical phase diffractions from (020), (110), (022) and (041) crystal planes[21]. Meanwhile, after adding SiO₂ and SiO₂-SO₃Li nanoparticles, the strong diffraction peaks of SiO₂/PVDF-HFP and SiO₂-SO₃Li/PVDF-HFP still exist, but the weak diffraction peaks disappear. Notably, the intensity of (110) crystal plane diffraction of SiO₂-SO₃Li/PVDF-HFP is significantly lower than that of PVDF-HFP, suggesting the crystallinity decreases while amorphous area increases. These changes have boosted the segmental motion of polymer chains and strengthened the movement of lithium-ion to some extent[22].

The rheological behavior of NNFQSE was investigated by rheometer. Typically, the storage modulus (G′) and loss modulus (G″) are usually used to characterize the fluidity of materials. Storage modulus (G′), also known as elastic modulus, refers to the amount of energy stored during elastic deformation of material. Loss modulus (G″), also known as viscous modulus, refers to the amount of energy lost during viscous deformation, reflecting the viscosity of the material. It is usually agreed that when G′ > G″, the materials will mechanically stiffen and exhibit shear-thickening property. On the contrary, when G″ > G′, the materials will turn soft and exhibit shear-thinning. As shown in Fig. 2f, at relatively low strain rate, G′ is obviously higher than G″, corresponding to the original morphology of quasi-solid electrolyte SiO₂-SO₃Li/PVDF-HFP before cycling (Supplementary Fig. 1b). G″ becomes larger gradually as the strain rate increases, revealing that the NNFQSE behaves shear thinning among the frequency region of 0.38 Hz-29.8 Hz. When the strain rate rises to a high value (>29.8 Hz), G′ exceeds G″ once again and the electrolyte demonstrates shear thickening to resist instantaneous impact. Furthermore, the shear-responsive process of NNFQSE can be signified in Fig. 2g. It can be explained that the hydrogen bonding between Si-OH and PVDF-HFP will be broken under the force slowly applied, manifesting as shear-thinning. At this stage, the NNFQSE will turn soft to absorb the stress caused by Li plating/stripping and cushion volumetric changes of Li anode. When subjected to a high-speed impact external force, the nanoparticles SiO₂-SO₃Li may instantly gather together to produce particle clusters, and the polymer molecules will snarl together, resulting in shear-thickening and rapid mechanically stiffen in response to inhibit the lithium dendrite growth.

## Properties of NNFQSE

Chemical composition and surface morphology are key factors in the wettability of polymer[23]. In this regard, the wettability of the composite polymers against liquid electrolyte (1 mol L⁻¹ LiTFSI/TEGDME) was evaluated by contact angle measurements. As depicted

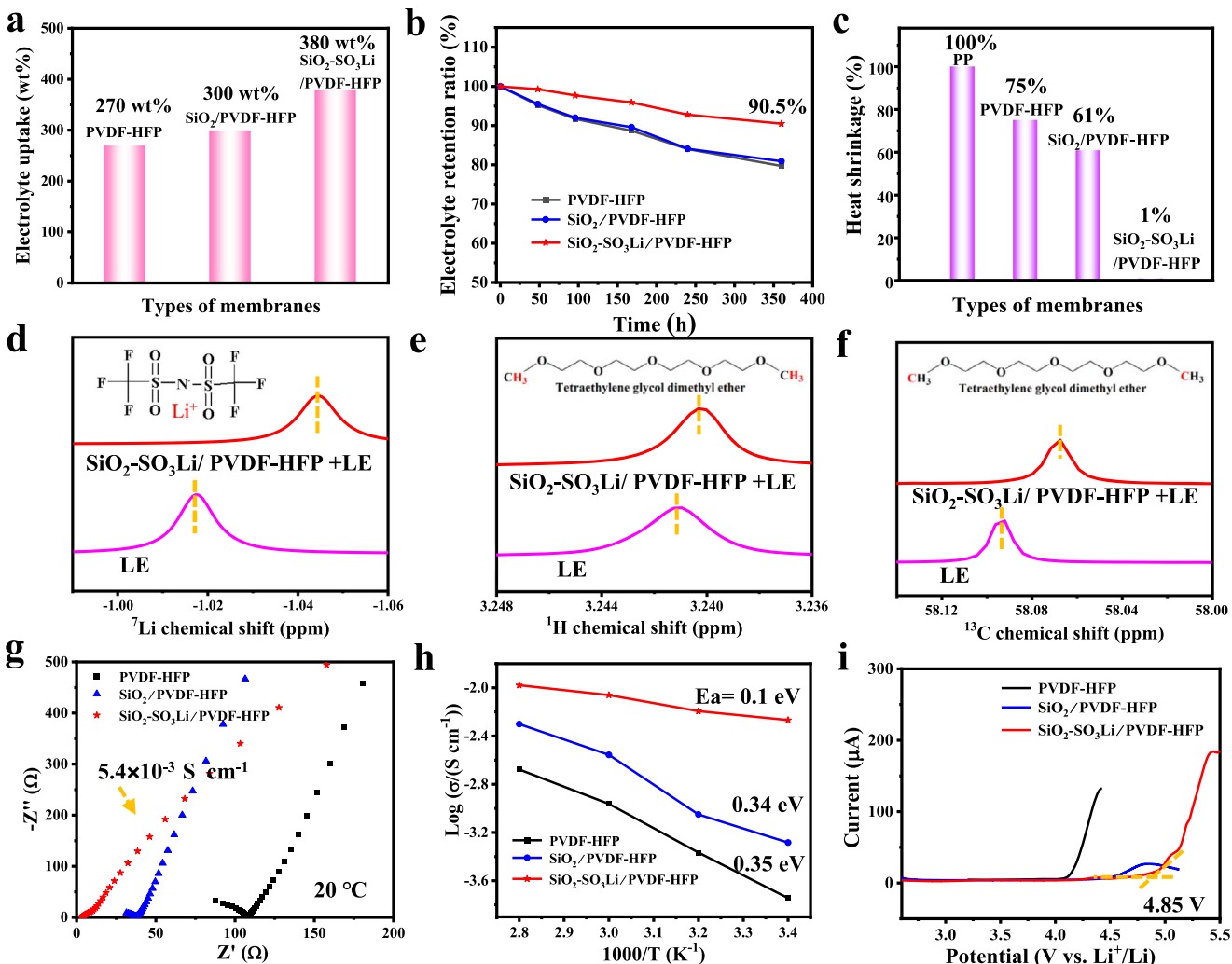

**Fig. 3 | Physical, chemical, and electrochemical performances of quasi-solid electrolytes. a** Electrolyte uptake; **b** Electrolyte retention ratio; **c** Heat shrinkage after heated at 180 °C for 1 h; **d–f** NMR analysis of the reaction between composite electrolyte and liquid electrolyte (LE); **g** EIS Nyquist plots at 20 °C; **h** The temperature dependency of ionic conductivity; **i** Liner sweep voltammograms at 25 °C with a scan rate of 1 mV s⁻¹.

in Supplementary Fig. 8, the SiO₂-SO₃Li/PVDF-HFP electrolyte behaves the smallest contact angle of 39.9°, while the bare PVDF-HFP electrolyte is 103.0° and SiO₂/PVDF-HFP electrolyte is 102.9°. Smaller contact angle indicates better electrolyte wettability[24]. Hence, the SiO₂-SO₃Li/ PVDF-HFP electrolyte with improved wettability can facilitate the infiltration of liquid electrolyte. In order to investigate the electrolyte preservation ability, the mass of these polymers before and after dipping in liquid electrolyte were measured, and the electrolyte uptakes were calculated through Eq. (1) shown in methods section. The electrolyte uptakes of PVDF-HFP, SiO₂/PVDF-HFP, and SiO₂-SO₃Li/PVDF-HFP are 270 wt%, 300 wt%, 380 wt% the dead weight, respectively (Fig. 3a). As a porous skeleton matrix material, PVDF-HFP with hexafluoropropylene chain segments possesses high elasticity and good swelling, contributing to adsorbing more electrolyte in a short time[25]. After adding porous fillers into PVDF-HFP, electrolyte retention ratio has been greatly increased shown in Fig. 3b. Significantly, the electrolyte retention ratio of SiO₂-SO₃Li/PVDF-HFP can still keep 90.5% in flowing air after 360 h, which is obviously superior to the others. The excellent electrolyte absorption and retention ratio of SiO₂-SO₃Li/PVDF-HFP may ensure the liquid electrolyte would not be consumed out during charge and discharge and thus increase the battery lifespan. What's more, the wrinkled porous nanofillers provide more effective contacts with liquid electrolyte, promoting Lewis acid-based interactions with anions and thus

dramatically boost the Li⁺ conductivity[20,26], as well as avoiding leakage issue to improve its safety[27].

Furthermore, heat shrinkage is another standard for good solid electrolytes. As shown in Fig. 3c, the SiO₂-SO₃Li/PVDF-HFP reveals hardly any heat shrinkage (about 1%) and maintains its shape stably after being heated at 180 °C for 1 h (Supplementary Fig. 9). In contrast, the traditional separator PP is dissolved completely, and the heat shrinkage of PVDF-HFP is about 75%, the SiO₂/PVDF-HFP is 61%, which demonstrates excellent stability and deformation resistance of NNFQSE SiO₂-SO₃Li/PVDF-HFP at high temperature.

The mentioned results suggest that there exists strong interaction among SiO₂-SO₃Li nanofillers, PVDF-HFP, and liquid electrolyte (LE). In order to give direct evidence, $^7$Li-NMR, $^1$H-NMR, $^{13}$C-NMR studies were performed, as shown in Fig. 3d–f. Based on comparison of the $^7$Li-NMR spectrums of LE, SiO₂-SO₃Li dipped in LE and SiO₂-SO₃Li/PVDF-HFP dipped in LE, the Li⁺ chemical environment has changed (Supplementary Fig. 10 and Fig. 3d). The $^7$Li resonances appear at −1.017 ppm in LE, − 1.034 ppm in SiO₂-SO₃Li + LE, −1.044 ppm in SiO₂-SO₃Li/PVDF-HFP + LE. The chemical shifts indicate that the SiO₂-SO₃Li and PVDF-HFP have both interacted with LiTFSI. The lone pair electrons of O on the surface of silica are negatively charged as electron donor group, which may attract Li⁺ of LiTFSI. Additionally, when the electron-absorbing sulfonic acid group is introduced on the surface of silica, the electron cloud density around the lithium nucleus will be reduced and

the chemical shift downfield. The -SO$_3$Li groups grafted on SiO$_2$ surface also has synergetic effect with LiTFSI. Therefore, the lithium salt LiTFSI of liquid electrolyte can be anchored inside SiO$_2$-SO$_3$Li nanospheres. The Li$^+$ chemical environment is also changed by adding the nanofillers into polymer skeleton PVDF-HFP, reflecting that the PVDF-HFP can also react with LiTFSI and trap LE to certain degree. Meanwhile, as shown in Fig. 3e, the $^1$H NMR signal at 3.241 ppm assigned to -CH$_3$ groups of the end of TEGDME chain has shifted to high field (right shift). Similarly, the changes of $^{13}$C-NMR have taken place in Fig. 3f. The results illustrate that the SiO$_2$-SO$_3$Li/PVDF-HFP has also put mutual effect with the solvent TEGDME of liquid electrolyte. Therefore, both lithium salt and solvent of liquid electrolyte can be easily stayed inside the quasi-solid electrolyte SiO$_2$-SO$_3$Li/PVDF-HFP, expressing high electrolyte retention as tested above.

The ionic conductivities of the quasi-solid electrolytes have been evaluated at different temperatures ranging from 20 °C–80 °C, and the results are calculated by the bulk resistance from electrochemical impedance spectra (EIS) (Fig. 3g, equation (2), Supplementary Fig. 11 and Fig. 3h). The Nyquist plots in Fig. 3g have shown that the SiO$_2$-SO$_3$Li/PVDF-HFP exhibits the lowest impedance and highest ionic conductivity $5.4 \times 10^{-3}$ S cm$^{-1}$ at 20 °C, indicating that the -SO$_3$Li groups on the surface of SiO$_2$ can provide extra binding sites for lithium ions and put on one more transfer path for Li$^+$ (Supplementary note 1). As shown in Fig. 3h, the temperature-dependent ionic conductivities of NNFQSE are fitted by the Arrhenius plots, and the activation energy can be calculated by $\sigma = A exp\left[\frac{-E_a}{RT}\right]$ equation (3), where σ is the ionic conductivity, A is the pre-exponential factor, E$_a$ is the activation energy, R is the ideal gas constant ($R = 8.314$ J mol$^{-1}$ K$^{-1}$) and T is the absolute temperature[28]. The SiO$_2$-SO$_3$Li/PVDF-HFP shows the lowest activation energy (0.1 eV), indicating the low energy barrier for Li$^+$ transport. Besides the superior ionic conductivity, the NNFQSE SiO$_2$-SO$_3$Li/PVDF-HFP has a wide electrochemical window up to 4.85 V, whereas the PVDF-HFP is oxidized at 4.1 V and SiO$_2$/PVDF-HFP at 4.5 V (Fig. 3i). As a consequence, it can prevent the occurrence of side reactions such as electrolyte decomposition, and ensure the battery safety furtherly.

## Application of NNFQSE in batteries

To verify the crucial role of designed NNFQSE in improving the electrochemical performance, long-term Li plating/stripping tests were performed on symmetric lithium batteries. Supplementary Fig. 12a shows the cycling performances of Li|SiO$_2$-SO$_3$Li/PVDF-HFP|Li, Li|SiO$_2$/PVDF-HFP|Li, and Li|PVDF-HFP|Li symmetrical cells at a current density of 0.5 mA cm$^{-2}$ at 25 °C, exhibiting the cell based on SiO$_2$-SO$_3$Li/PVDF-HFP behaves the smallest polarization voltage about 25 mV. Moreover, when the inner liquid electrolyte of NNFQSE is replaced from 1 mol L$^{-1}$ LiTFSI/TEGDME to LiPF$_6$/EC-DEC (Supplementary Fig. 12b), the cell Li|SiO$_2$-SO$_3$Li/PVDF-HFP|Li still runs the most stable over 800 h with the lowest polarization voltage about 25 mV, compared to those of the PVDF-HFP and SiO$_2$/PVDF-HFP, which are about 78 mV and 50 mV, respectively. These results further provide compelling evidence that the -SO$_3$Li groups can effectively promote Li$^+$ transport and improve the battery lifetime. Remarkably, a long cycling stability over 2000 h with extremely low polarization voltage (about 50 mV) at a relatively high current density of 1 mA cm$^{-2}$ can be achieved via NNFQSE SiO$_2$-SO$_3$Li/PVDF-HFP (Fig. 4a), suggesting effective inhibition of lithium dendritic formation. The substantial improvement in electrode cycling performance may be ascribed to the shear-thinning and shear-thickening properties of NNFQSE SiO$_2$-SO$_3$Li/PVDF-HFP, which release the stress during Li plating/stripping and suppress nonuniform Li deposition and dendrite growth.

In light of the high electrolyte retention, good thermal tolerance, high ionic conductivity, electrochemical stability, and lithium dendritic inhibition of NNFQSE SiO$_2$-SO$_3$Li/PVDF-HFP, we employed the NNFQSE to assemble Li-O$_2$ batteries to demonstrate its practical application ability. Lithium metal was assembled as the anode, and

active carbonaceous material loaded 3 wt% Ru catalyst was used as the cathode. The cycle performance of Li-O$_2$ batteries was checked through galvanostatic discharge-charge at 100 mA g$^{-1}$ with a limited capacity of 600 mAh g$^{-1}$ at 25 °C. It can be clearly seen from Fig. 4b, c that the Li-O$_2$ cell with NNFQSE SiO$_2$-SO$_3$Li/PVDF-HFP achieves long life above half a year (>5000 h) without any lithium dendrites, which is several times as the conventional porous separator PP, PVDF-HFP and SiO$_2$/PVDF-HFP. Typical charge/discharge curves for the Li/ SiO$_2$-SO$_3$Li/ PVDF-HFP/O$_2$ battery is plotted in Fig. 4d, demonstrating reversible and stable cycling processes. Figure 4e shows the rate capability of Li|SiO$_2$-SO$_3$Li/PVDF-HFP|O$_2$ at different specific currents at 25 °C. The battery displays excellent rate performances at the cycling rates of 150 mA g$^{-1}$, 200 mA g$^{-1}$, 500 mA g$^{-1}$, 1000 mA g$^{-1}$, 150 mA g$^{-1}$, 200 mA g$^{-1}$. When the cycling rate is switched back to 150 mA g$^{-1}$ and 200 mA g$^{-1}$, the cell exhibits a recovery of its voltage and keeps stable. By contrast, Li-O$_2$ cells with PVDF-HFP (Supplementary Fig. 13a) and SiO$_2$/PVDF-HFP (Supplementary Fig. 13b) deliver unstable rate performance at different specific currents. The main reason can be imagined as that the NNFQSE SiO$_2$-SO$_3$Li/PVDF-HFP exhibits shear-thickening in response to instantaneous impact of macroscopic strain and thus restrain the dendrites growth at high rates. Besides, the EIS curves for the Li-O$_2$ cells at room temperature also demonstrate that NNFQSE SiO$_2$-SO$_3$Li/PVDF-HFP performs lower total impedance and well interfacial contact with electrode (Fig. 4f). It is interesting to note that the total impedance only increases from 91 Ω to 330 Ω after cycling (Supplementary Fig. 14). The dramatic boost in the lifespan of Li-O$_2$ battery can be attributed to the high electrolyte retention and rheological behavior of NNFQSE SiO$_2$-SO$_3$Li/PVDF-HFP.

To confirm this, the morphology evolution of NNFQSE SiO$_2$-SO$_3$Li/PVDF-HFP after 5000 h Li-O$_2$ cell cycling was conducted. It is worth noting that the NNFQSE SiO$_2$-SO$_3$Li/PVDF-HFP changes from transparent (Supplementary Fig. 1b) to elastic jelly (Supplementary Fig. 15a) after 5000 h circulation, and remains moist interior, which is visible to the naked eyes. In stark contrast, the SiO$_2$/PVDF-HFP and pure PVDF-HFP completely dried out or even broke (Supplementary Fig. 15b, c). The phenomenon intuitively proves that the sulfonic modified electrolyte is equipped with strong liquid electrolyte preserving performance and forms non-Newtonian fluid under the shear of macroscopic strain and electric field force in the plating/stripping process. Furthermore, Supplementary Fig. 16 shows the surface SEM analysis and EDS elemental mapping of NNFQSE SiO$_2$-SO$_3$Li/PVDF-HFP after 5000 h cycling, demonstrating that there still maintain rough honeycomb morphology channels for Li-ion transport (Supplementary Fig. 16a) and the SiO$_2$-SO$_3$Li nanofillers inside remain original wrinkled radial structure (Supplementary Fig. 16b). The elements of C, F, Si, O, S are uniformly distributed that can be clearly seen in elements mapping (Supplementary Fig. 16c). Meanwhile, its cross-profile morphology also remains well with many vertical channels (Fig. 5a) and so does nanofillers (Fig. 5b). Its elemental mapping in cross section distributes uniformly too (Supplementary Fig. 17). Besides, the chemical bonding states of SiO$_2$-SO$_3$Li/PVDF-HFP with liquid electrolyte inside are noted by FTIR spectrum before and after cycling (Supplementary Fig. 18, Fig. 5c). Before cycling, the peaks at 1192 cm$^{-1}$ and 1093 cm$^{-1}$ assigned to -SO$_3$ stretching and Si−O−Si stretching adsorption respectively experience a 2 cm$^{-1}$ blue shift and 7 cm$^{-1}$ blue shift when compared to Fig. 2d, revealing the interaction between SiO$_2$-SO$_3$Li and liquid electrolyte. And after long cycling, these peaks did not turn shift, and there no new vibration peaks generate, which confirms the stability of the designed electrolyte. Moreover, there also no new functional groups appear in the XPS spectra (Fig. 5d–f) after 5000 h cycling, verifying its stability furtherly. In addition, as shown in Supplementary Fig. 19, the peaks around 688 eV and 689.1 eV are attributed to -CF$_3$ and -CF$_2$ respectively, which are typical signals for PVDF-HFP[29]. After absorbing liquid electrolyte, Li$^+$ of -SO$_3$Li groups may react with F$^-$ of LiTFSI to generate LiF at 685 eV, while the area of -CF$_3$ is reduced and -CF$_2$ area

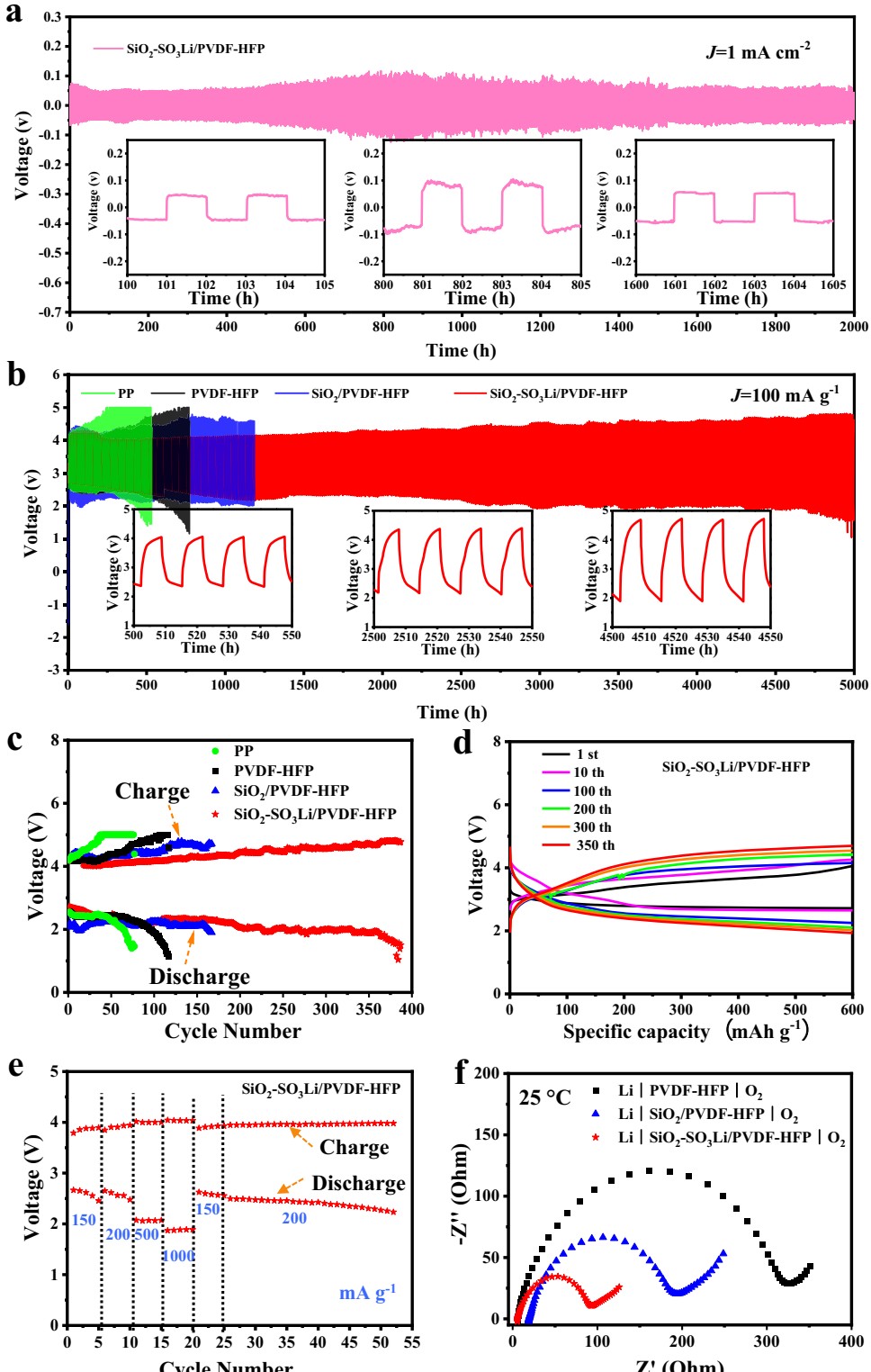

**Fig. 4 | Electrochemical performance of NNFQSE. a** Symmetric lithium battery based on SiO$_2$-SO$_3$Li/PVDF-HFP tested at 1 mA cm$^{-2}$, 1 mAh cm$^{-2}$ at 25 °C, the inserts are selected voltage profiles for a different time; **b, c** Cycling stability of Li-O$_2$ cells with three kinds of quasi-solid electrolytes and conventional separator (PP) at 100 mA g$^{-1}$ with a limited capacity of 600 mAh g$^{-1}$; **d** Typical charge/discharge profiles of the Li-O$_2$ cell with SiO$_2$-SO$_3$Li /PVDF-HFP at a specific current of 100 mA g$^{-1}$; **e** Rate performance of Li|SiO$_2$-SO$_3$Li /PVDF-HFP |O$_2$; **f** EIS curves for the full cells at room temperature.

increased (Fig. 5d). Notably, after cycling, the area of LiF is greatly increased owing to the interaction between lithium metal and -CF$_3$ of LiTFSI and PVDF-HFP, coupling with the same changes of -CF$_3$ and -CF$_2$ as before. Moreover, the peaks assigned to LiF, -CF$_3$, and -CF$_2$ bonds move +0.5 eV, +0.7 eV, and +0.4 eV, respectively, confirming the presence of a strong electron-withdrawing group -SO$_3$H around F$^-$, and there exists chemical forces between them[29–33]. Figure 5e exhibits the S 2$p$ spectra, the peak around 167.3 eV is ascribed to the characteristic peaks of -SO$_3$Li, which attests the -SO$_3$Li functional groups have been successfully introduced on SiO$_2$[34–36]. The peaks 169 eV and 170.1 eV are

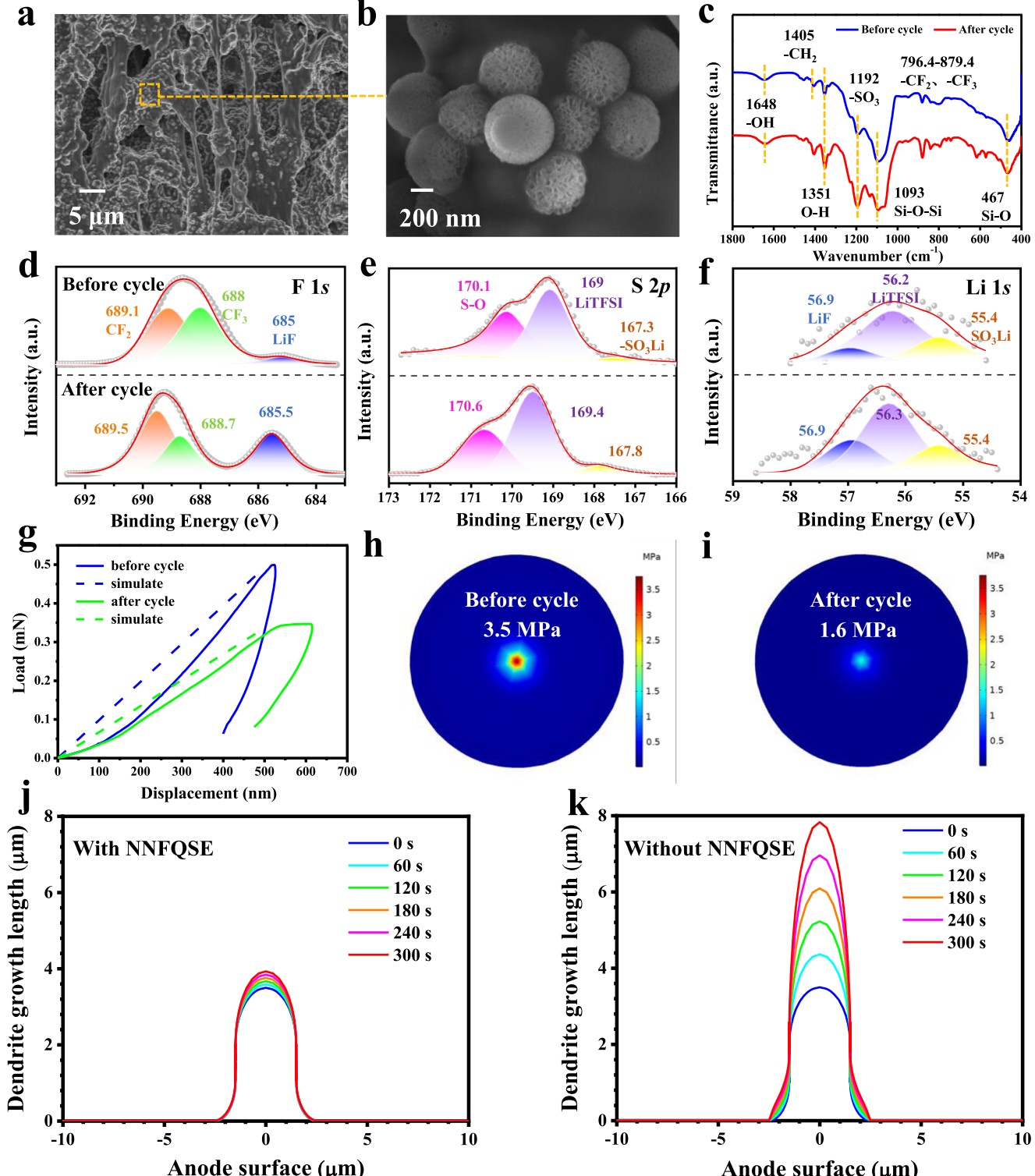

**Fig. 5 | Characterization of NNFQSE SiO₂-SO₃Li/PVDF-HFP before and after 5000 h cycling. a** Cross-section SEM images; **b** enlarged view of **a**; **c** Infrared spectrum among the region of 400–1800 cm⁻¹; XPS analysis: **d** F 1s; **e** S 2p; **f** Li 1s; **g** Nano-indentation test and simulated load-displacement curve. The horizontal coordinate corresponds to lithium dendrite length, and the vertical coordinate corresponds to the force; **h**, **i** COMSOL simulation of the electrolyte deformation. The pressure cloud diagram under the same deformation before **h** and after **i** cycling; COMSOL simulations of Li dendrite growth during prolonged time of 300 s **j** with NNFQSE or **k** without NNFQSE.

attributed to LiTFSI and S-O bonds. The binding energy also shifted left after cycling[37,38]. From Fig. 5f, there is a weak peak in Li 1s spectrum assigned to -SO₃Li at 55.4 eV[39]. Peaks at 56.2 eV corresponds to LiTFSI and 56.9 eV assigns to LiF[40]. LiF, an electrical insulator, can stop electrons from leaking through the SEI, preventing the electrolytes'

ongoing deterioration and capacity loss. And Li ions will not tend to vertical dendrite growth due to the high interfacial energy of LiF to Li metal[41].

It is worth noting that the lithium anode surface of Li|SiO₂-SO₃Li/ PVDF-HFP|O₂ exhibits smooth without obvious dendrite growth after

long-term cycling (see Supplementary Fig. 20 and Supplementary Note 2), which may also benefit from the mechanical properties of NNFQSE SiO$_2$-SO$_3$Li/PVDF-HFP. In order to prove this, nano-indentation tests were carried out before and after cycling for 200 h. The hardness and elastic modulus respectively decrease form 93.19 MPa and 2.27 GPa to 49.6 MPa and 1.02 GPa after circulation, indicating that the NNFQSE SiO$_2$-SO$_3$Li/PVDF-HFP exhibits shear-thinning under the stress and turns softer than before. It can also be seen from the load-displacement curve of nano-indentation (solid line in Fig. 5g) that the NNFQSE is suffered less stress with same displacement under the pressure of nano-indenter after cycling, which demonstrates that the NNFQSE can absorb the stress caused by Li plating/stripping. In addition, mechanical simulations were taken to further study the resistance of the NNFQSE SiO$_2$-SO$_3$Li/PVDF-HFP to dendrite growth using COMSOL Multiphysics. A simulate electrolyte membrane with diameter of 19 mm and a thickness of 100 μm was considered. A prescribed displacement, which corresponds to the dendrite growth, was applied at the center of the electrolyte membrane with monotonic increments of 5 nm for 100 steps. The final deformation was amplified 800 times, as shown in Supplementary Fig. 21. The load-displacement curve shows that the ultimate load becomes lower after circulation with identical dendrite growth, which is consistent with the results of nano-indentation test (dash line in Fig. 5g). Besides, the contours of Von-Mises stress presented in Fig. 5h–i reveals that the maximum Von-Mises stress after cycling is much lower than before, confirming that the membrane has become softer and eliminates the stress after cycling, then less likely to be punctured by uncontrollable lithium dendrites[42]. On the other hand, in areas where the lithium deposition is uneven, or the applied current density is of a large magnitude, the sudden increase of contact stress will make the quasi-solid electrolyte turn to shear-thickening. At such state, the stiffened NNFQSE can hardly be deformed even forced by aggressive stress, and thus suppress the dendrite growth mechanically. As can be seen in Fig. 5j–k, with a same violent applied current density and the same initial protrusion on anode, the dendrite with the restriction of the stiffened NNFQSE increased from 3.5 μm to 3.9 μm within 300 s, the growth rate is only ~11%. While the one without NNFQSE increased to 7.8 μm, showing more than ten times growth rate of ~123%. Therefore, with NNFQSE, the ionic flux will be more slight and the dendrite growth is also perfectly inhibited (Supplementary Fig. 22). These results above reveal that the NNFQSE SiO$_2$-SO$_3$Li/PVDF-HFP is of great significance to the realization of long life and high safety Li-O$_2$ batteries without lithium dendrite growth and penetration. Furthermore, on the cathode side, the NNFQSE SiO$_2$-SO$_3$Li/PVDF-HFP not only exhibits robust chemical stability against products during cycling, but also traps the liquid electrolyte inside to prevent the side reaction with cathode and reduce the residue of solid-state products, which further ensure the Li-O$_2$ batteries' life (Supplementary Figs. 23, 24 and Supplementary Note 2).

In summary, a notion of non-Newtonian fluid for realizing long life and dendrite-free Li-O$_2$ battery has been designed by NNFQSE SiO$_2$-SO$_3$Li/PVDF-HFP. Profiting from its honeycomb cross-linked network, wrinkled nanofillers, and component interactions, the NNFQSE SiO$_2$-SO$_3$Li/PVDF-HFP can be endowed high liquid electrolyte retention in semi-open system, high Li$^+$ conductivity, low activation energy, wide electrochemical window, and good thermal tolerance. Revealed by the combination of theoretical mechanical simulations and nano-indentation tests, the NNFQSE exhibits shear-thinning to eliminate the strain under long-term Li plating/stripping. Furthermore, the COMSOL simulations also demonstrate shear-thickening in response to instantaneous impact of macroscopic strain and restrain the dendrites growth. As a result, the NNFQSE SiO$_2$-SO$_3$Li/PVDF-HFP enables dendrite-free lithium plating/stripping in symmetrical batteries for 2000 h at 1 mA cm$^{-2}$ with low polarization voltage ~50 mV, and realizes long life Li-O$_2$ batteries over 5000 h under 100 mA g$^{-1}$. In general, the NNFQSE opens an avenue for Li metal batteries to resolve the long-lasting dendrite growth and enhance the electrochemical performance.

## Methods

### Preparation of modified nanomaterials

The modified nanoparticles SiO$_2$-SO$_3$Li were prepared by following three steps. Firstly, porous SiO$_2$ nanoparticles were synthesized as follows[43]: 2.0 g hexadecylpyridinium bromide (CPB) and 1.2 g carbamide were dissolved in 60 mL deionized water. This mixture was marked as A. 3 mL pentanol, 5.4 mL tetraethyl orthosilicate (TEOS), and 60 mL cyclohexane were mixed and stirred strongly at room temperature. This mixture was marked as B. A was stirred intensely until the milky solution turned transparent, and then B was added to A by drops. The mixture was continued stirring for 1 h after it turned into milky solution. Subsequently, the obtained solution was transferred into Teflon-lined stainless-steel autoclave to heat at 120 °C for 24 h. After that, the material was filtered and washed with deionized water several times, then dried in a vacuum oven at 70 °C for 6 h. The acquired powders were placed into muffle furnace and calcined at 600 °C for 5 h to remove the surfactant. As a result, the white powders of SiO$_2$ nanoparticles were obtained. Secondly, the SiO$_2$ nanoparticles were dipped in excess H$_2$SO$_4$ for 24 h, then separated by centrifuge, washed three times by deionized water, and dried under vacuum at 70 °C, then sulfated silica nanoparticles (SiO$_2$-SO$_3$H) were synthesized successfully. Lastly, the SiO$_2$-SO$_3$H materials were dipped into enough LiOH for 24 h. SiO$_2$-SO$_3$Li nanoparticles were also obtained after centrifuging, washing, and drying.

### Preparation of PVDF-HFP-based polymer electrolytes

Three kinds of PVDF-HFP based polymer electrolytes were synthesized via tape casting process. For the first electrolyte, 0.25 g PVDF-HFP was dissolved in 4.5 g acetone. Then 0.25 g ethyl alcohol was added into the above solution by dropwise and stirred for 0.5 h. The obtained transparent solution was casted onto a rectangular Teflon mold and natural drying[44]. After drying, the membrane was gently stripped off and dried furtherly at 60 °C in a vacuum. This electrolyte was named PVDF/HFP. For the second electrolyte, the synthesis process was similar. 0.139 g PVDF-HFP was dissolved in 4.5 g acetone and stirred 1 h until PVDF-HFP dissolved completely. Then 0.110 g SiO$_2$ nanoparticles were added to the above mixture. The rest operations were the same as the synthesis of PVDF/HFP electrolyte. This one was marked as SiO$_2$/PVDF-HFP. For the third electrolyte, 0.110 g SiO$_2$ were replaced by 0.110 g SiO$_2$-SO$_3$Li. This composite electrolyte was marked as SiO$_2$-SO$_3$Li/PVDF-HFP with 100 μm thick and 19 mm in diameter. Then the synthesized electrolytes were dipped in conventional liquid electrolyte (1 mol L$^{-1}$ LiTFSI/TEGDME) for 24 h in argon-filled glove box. Before assembling batteries, the extra liquid electrolyte on the surface was removed by filter paper and the polymer electrolyte turned transparent with semi-dry surface (Supplementary Fig. 1b). Moreover, the polymer electrolyte SiO$_2$-SO$_3$Li/PVDF-HFP was flexible after bended several times (Supplementary Fig. 1c).

### Preparation of the electrode

The lithium metal anode is 0.6 mm in thickness and 16 mm in diameter. The carbon materials of cathode were homemade, which behave hollow fiber pipelines to allow oxygen in and out. The catalyst RuCl$_3$ solution (5 mg mL$^{-1}$ Ru$^{3+}$) was added to the above carbon materials. After vacuum freeze-drying treatment, the materials were put into a tube furnace and reduced at 1000 °C under a hydrogen atmosphere for 2 h. Then the cathode was prepared successfully with 1 mm in thickness and 6 mm in diameter and the mass was ~21.2 mg cm$^2$. The loading of Ru nanoparticles in cathode was 3 wt%, which was measured by inductively coupled plasma-atomic emission spectrometry (ICP-AES, Leeman Labs Inc., USA).

## Characterizations and measurements

The resultant materials were characterized using X-ray diffraction (XRD), X-ray photoelectron spectroscopy (XPS), Fourier Transform Infrared Spectroscopy (FTIR), R/S Rheometer (Brookfield), scanning electron microscopy (SEM), and transmission electron microscopy (TEM). The XRD experiment was conducted on a Bruker D8 Advance diffractometer with Cu Ka radiation and generated at 40 kV and 40 mA. The XPS measurement was performed on an ESCALAB Xi + (Thermo Scientific) spectrometer equipped with an Al Kα achromatic X-ray source (spot 650 microns). FTIR (Tensor 27, Bruke Germany) measurement was conducted ranging 4000–400 cm$^{-1}$. The rheometer test used the modulus oscillation mode at 0.1-100 Hz. The SEM observation was performed on Hitachi UHR field emission SEM (SU8220) and TEM images was recorded on a JEM-2100HR TEM (JEOL, Japan). Surface tension & surface contact angle tester was obtained by OCA40 Micro instrument from Dataphysics Company in Germany. Nuclear Magnetic Resonance (NMR) investigation, $^{7}$Li, $^{1}$H, $^{13}$C experiments were carried out on Bruker 600 MHz Bruker.

The bulk resistances were determined from the electrochemical impedance spectroscopy (EIS) tested by AutoLab workstation (Metrohm, Autolab B.V., PGSTAT30), and the frequency range from10$^{6}$ Hz to 0.1 Hz with an AC amplitude of 10 mV. Then calculated the ionic conductivity (σ) using a general formula $\sigma = \frac{L}{RS}$ equation (2). The resistance (R) obtained from impedance spectrum; the thickness of electrolyte was obtained by micrometer (L); the effective contact area (S) was the section between electrolyte and blocking electrode. The electrochemical stability window was investigated by linear sweep voltammetry (LSV) (2.5 to 6 V) measurement performed on placing the electrolyte between a lithium metal and a stainless steel sheet at 1.0 mV s$^{-1}$. The lithium symmetrical batteries and Li-O$_2$ batteries were assembled in an Ar-filled glove box (O$_2$ < 0.01 ppm, H$_2$O < 0.01 ppm) with CR2032 coin cells and tested by NEWARE testing system (CT-4008-5 V 10 mA-164) at room temperature.

## Electrolyte uptake and retention test

The three kinds of polymer electrolytes were dipped in conventional liquid electrolyte 1 mol L$^{-1}$ LiTFSI/TEGDME in glove box respectively, then wiped the excess liquid electrolyte gently with filter paper after reaching saturation point. Afterwards, weighing the quality of the polymer electrolyte before (W$_0$) and after (W) absorbing liquid electrolyte. The following formula can be used to calculate electrolyte uptake (EU):

$$EU = (W - W_0)/W_0 \times 100\%. \qquad (1)$$

Then weight the mass of composite electrolyte at different periods of time to get the data of liquid retention.

## Data availability

The data that support the findings of this study are available within the article and its Supplementary Information files. All other relevant data supporting the findings of this study are available from the corresponding authors upon reasonable request. Source data are provided with this paper.

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

## Acknowledgements

This work was supported by the National Science Foundation of China (no. 21776105 to H.S.), the Natural Science Foundation of Guangdong Province (no. 2019A1515011720 to H.S.), Science and Technology Program of Guangzhou (no. 201904010340 to H.S.).

## Author contributions

H.S. and G.Z. conceived the idea and co-wrote the manuscript. Z.C., L.D., and Z.L. participated in the initial discussions of the project. G.Z. and X.Z. prepared the samples and performed the experiments. Y.H. and J.W. performed mechanical simulations. G.Z., H.S., and T.Y. were involved in the analysis of the experimental data.

## Competing interests

The authors declare no competing interests.
