## [Peer Review File · Nature Communications]

A non-Newtonian fluid quasi-solid electrolyte designed for long life and high safety Li-O₂ batteriesREVIEWER COMMENTS

Reviewer #1 (Remarks to the Author):

A novel non-Newtonian fluid quasi-solid electrolyte (NNFQSE) SiO₂-SO₃Li/PVDF-HFP is proposed in this work to address the major issue of anode (Li dendrite growth) and the inherent issue of Li-O₂ battery as its semi-open structure (volatilization and leakage of liquid electrolytes). As a result, the lithium symmetrical battery can operate for more than 2000 hours at 1 mA cm⁻² at room temperature, while the quasi-solid Li-O₂ battery has a superior long life of more than 0.5 years (> 5000 h at 100 mA g⁻¹). This is a very shocking result in the field of lithium-air batteries, especially in terms of stability. This suggests new ideas for the design of long-life lithium-air batteries.

However, there are some issues in this work that require further discussion.

- (1) What causes such high Li⁺ conductivity at room temperature? As presented, the 5.4×10⁻³ S/cm of quasi-solid electrolyte is higher than that of liquid electrolyte, such as, LiTFSI/TEGDME;
- (2) The physical properties of the quasi-solid electrolyte, especially its shear thickened property should be discussed. And the inhibition mechanism and inhibition effect of lithium dendrites should be clarified;
- (3) The residue of solid-state products in cathode and the pulverization and dendrite situation of lithium anode after long-term stability tests should be further discussed.

Reviewer #2 (Remarks to the Author):

In this work, the authors reported that using SiO₂-SO₃Li/PVDF-HFP as non-Newtonian fluid quasi-solid electrolyte in Li-O₂ battery can effectively eliminate dendrite issues during cycling measurements along with good liquid electrolyte retention and Li⁺ conductivity. As the results, the battery can perform excellent cycling stability of 5000 hours at 100 mA g⁻¹. Some issues are addressed as follows:

1. It is really hard to see that the S are uniformly distributed on SiO₂-SO₃Li particles from EDS-mapping result in Figure S3. Please check the result and give more clear demonstration.
2. The FTIR results should give more clarifications between the SiO₂, SiO₂-SO₃H and SiO₂-SO₃Li to prove the varied surface chemistry. It is claimed by author that the peak of Si-OH in 2347 cm⁻¹ disappears in SiO₂-SO₃Li compared to SiO₂ due to sulfonated treatment, but why there is a peak of Si-OH at 1632 cm⁻¹ in both SiO₂ and SiO₂-SO₃Li samples in Figure 2d and Figure S5? What the difference between these vibration features?
3. Please explain why the core level XPS spectra of S 2p seems very poor signal but showed much stronger intensity in survey spectra in Figure 2e.
4. The porosity of control-group samples should be provided to prove the superior microstructure properties of skeleton matrix materials.
5. Please explain why the stable electrochemical window of SiO₂-SO₃Li/PVDF-HFP is much higher than the others? Besides, LSV with same scan rate to low potential area should also provided to examine its reduction behavior.
6. The sample of SiO₂ /PVDF-HFP showed relatively lower voltage hysteresis compared to SiO₂-SO₃Li/PVDF-HFP. In Figure 4b,c If reaction kinetics of Li⁺ in SiO₂-SO₃Li/PVDF-HFP is higher than other, why this phenomenon occurred?

Reviewer #3 (Remarks to the Author):

This work reported a composite quasi-solid electrolyte SiO₂-SO₃Li/PVDF-HFP with non-Newtonian fluid properties for Li-O₂ batteries (LOBs). Intensive mechanical and electrochemical characterizations were conducted to investigate the properties of the designed electrolyte. The

assembled LOBs exhibited an ultra-long cycling life. But the current manuscript lacks analyses and discussions regarding battery chemistry and anode/electrolyte/cathode interfaces. The promotion effect was oversimplified by component interactions. There are also some problems to be clarified before the final acceptance.

1. The author calculated the contact angle of SiO₂-SO₃Li/PVDF-HFP with a different tangent center from control samples (Figure S7). A wrong conclusion about wettability was obtained. From the direct view, the contact angles of the three samples are similar. The enhanced wettability of SiO₂-SO₃Li/PVDF-HFP is doubting.
2. The order of Figures in supporting information is inconsistent with the manuscript.
3. The preparation procedure of the cathode should be illustrated in the experimental section.
4. From presented photos of SiO₂-SO₃Li/PVDF-HFP before and after cycling, the cycled electrode became thicker than the initial, and why?
5. From a recent literature review of LOBs (doi.org/10.1039/D2CS00003B), the battery chemistry is significantly affected by the electrolyte. How the SiO₂-SO₃Li/PVDF-HFP electrolyte affects the growth and decomposition of discharge products is yet lacking, as well as the crystal structure and morphology.
6. Although SiO₂-SO₃Li/PVDF-HFP exhibited a wide electrochemical window, reactive oxygen species generated during the discharging and charging of LOBs could also induce the decomposition of the electrolyte and functional groups, especially during ultra-long operation. Relevant discussions and analyses are recommended in order to obtain a more comprehensive understanding of the stability of the designed electrolyte.
7. As shown in Figure S14, the impedance of the cycled LOB increased significantly. This could be caused by the coverage of undecomposed discharge products or the degradation/evaporation of the electrolyte. The actual reasons should be clarified.
8. From Figure 2c and Figure S2c, similar morphology was observed on SiO₂-SO₃Li and SiO₂. How did the author define the "surface channels" and conclude the "deeper and richer" surface channels of SiO₂-SO₃Li than SiO₂?
9. The introduction of the -SO₃Li group significantly changed the morphology of composite polymers, enabling uniform distribution of SiO₂ and providing Li⁺ diffusion channels. But the extra roles of the -SO₃Li groups in promoting battery performance are ambiguous and lack compelling evidence.

Point-by-point Response to the Reviewers

REVIEWER COMMENTS

Reviewer #1 (Remarks to the Author):

A novel non-Newtonian fluid quasi-solid electrolyte (NNFQSE) SiO₂-SO₃Li/PVDF-HFP is proposed in this work to address the major issue of anode (Li dendrite growth) and the inherent issue of Li-O₂ battery as its semi-open structure (volatilization and leakage of liquid electrolytes). As a result, the lithium symmetrical battery can operate for more than 2000 hours at 1 mA cm⁻² at room temperature, while the quasi-solid Li-O₂ battery has a superior long life of more than 0.5 years (> 5000 h at 100 mA g⁻¹). This is a very shocking result in the field of lithium-air batteries, especially in terms of stability. This suggests new ideas for the design of long-life lithium-air batteries. However, there are some issues in this work that require further discussion.

Response: We thank the reviewer for the positive comments on the impact of our work. We have revised the manuscript and answered questions following your comments.

1. What causes such high Li⁺ conductivity at room temperature? As presented, the 5.4×10⁻³ S/cm of quasi-solid electrolyte is higher than that of liquid electrolyte, such as, LiTFSI/TEGDME.

Response: Thanks for the valuable comment. The high Li⁺ conductivity of quasi-solid electrolyte SiO₂-SO₃Li/PVDF-HFP at room temperature can be mainly attributed to the plenty of liquid electrolyte LiTFSI/TEGDME inside and Li⁺-conductive groups -SO₃Li.

- 1) The electrolyte uptake of SiO₂-SO₃Li/PVDF-HFP reaches 380 wt% the dead weight, which is dominant the Li⁺ conductivity.
- 2) Previous reports demonstrated that the sulfonate groups -SO₃Li with electronegativity can simultaneously facilitate access by the electrolyte to promote ion pair dissociation and increase the mobility of Li⁺. (*Adv. Mater.* 2021, 33, 2105178) Furthermore, -SO₃Li sites also provide the lithium ion hopping pathways to improve Li⁺ conductivity. (*Nano Lett.* 2021, 21, 2997–3006; *J. Am. Chem. Soc.* 2019, 141, 5880–5885)

Editorial Note: Figure below reproduced from Qiao, L., Oteo, U., Martinez-Ibañez, M. et al. Stable non-corrosive sulfonimide salt for 4-V-class lithium metal batteries. *Nat. Mater.* **21**, 455–462 (2022), with permission from Springer Nature.

3) The polymer skeleton PVDF-HFP provides a certain amount of Li^+ driving force through its segmental motion.

Therefore, the NNFQSE $\text{SiO}_2\text{-SO}_3\text{Li/PVDF-HFP}$ with semi-dry surface can be seen as a “liquid electrolyte” to some extent, which exhibits high ion conductivity $5.4 \times 10^{-3} \text{ S/cm}$ at room temperature, equivalent to the reported conventional liquid electrolyte LiTFSI/TEGDME . (*Nat. Mater.* 2022,21, 455–462) (Fig. R1).

Fig. R1 Ionic conductivity of conventional liquid electrolytes at 20 °C reported in reference *Nat. Mater.* 2022,21, 455–462.

Updates to the revised manuscript and supplementary information: We have added these detail discussions in the revised supplementary information (supplementary Note 1, page S7) and mentioned in the revised manuscript on Page 12, highlighted in yellow.

2. The physical properties of the quasi-solid electrolyte, especially its shear thickened property should be discussed. And the inhibition mechanism and inhibition effect of lithium dendrites should be clarified;

Response: Thanks for the valuable suggestion. As shown in Fig. 2f-g, the quasi-solid electrolyte exhibits a double shear-thickening behavior at initial shear rates and high shear rates respectively, and its intermediate state behaves shear-thinning. The special rheological properties reveal that the hydrogen-bonding between Si-OH and PVDF-HFP will be broken under the force slowly applied, which leads to the loss modulus (G'') become larger than storage modulus (G'), displaying shear-thinning behavior. When subjected to a high-speed impact external force, the nanoparticles $\text{SiO}_2\text{-SO}_3\text{Li}$

instantly gather together to produce particle clusters and the polymer molecules will snarl together, resulting G' exceeds G'' once again, namely shear-thickening. (*Compos. Struct.* 2021,266, 113806).

To clarified the inhibition mechanism and inhibition effect of lithium dendrites, we have carried out the COMSOL Multiphysics. On the one hand, when the NNFQSE is in the state of shear-thinning, its mechanical stiffness will become more and more soft during the charging/discharging cycles, which effectively eliminates the stress caused by the develop of dendrite (see Fig. 5h-i). On the other hand, in areas where the lithium deposition is uneven, or the applied current density is of a large magnitude, the sudden increase of contact stress will make the quasi-solid electrolyte turn to shear-thickening. At such state, the stiffened NNFQSE can hardly be deformed even forced by aggressive stress, and thus suppress the dendrite growth mechanically. As can be seen in Fig. R2, with a same violent applied current density and a same initial protrusion on anode, the dendrite with the restriction of the stiffened NNFQSE increased from 3.5 μm , to 3.9 μm , within 300 s, the growth rate is only about 11% (Fig. R2a). While the one without NNFQSE increased to 7.8 μm , showing more than ten times growth rate of about 123% (Fig. R2b). Therefore, with NNFQSE, the ionic flux will be more slight and the dendrite growth is also perfectly inhibited (Fig. R3).

Fig. R2 COMSOL simulations of Li dendrite growth during prolonged time of 300 s (a) with NNFQSE or (b) without NNFQSE.

Fig. R3 Ionic flux and Li dendrite growth simulations during prolonged time of 300 s (a-c) with NNFQSE or (d-f) without NNFQSE.

Updates to the revised manuscript and supplementary information: We have added these COMSOL simulations in Fig. 5(j-k) and supplementary Fig. 22, and detail discussion in the revised manuscript on Page 8 (Line156-163) and Page 21 (Line 383-393), highlighted in yellow.

3. The residue of solid-state products in cathode and the pulverization and dendrite situation of lithium anode after long-term stability tests should be further discussed.

Response: Thanks for the comments. The characterizations of cathode and lithium anode after long-term cycling have performed by high resolution scanning electron microscope (SEM). As shown in Fig. R4a, the cathode of Li | SiO₂-SO₃Li/PVDF-HFP | O₂ still maintains hollow tubular structure and there only exists little residue of solid-state products after 5000 h cycling. In contrast, the cathode structure of control groups becomes deformation or even collapsed, and there also filled with residue, which hinders the persistent cycling (Fig. R4b-c). On the anode side, the lithium anode surface of Li | SiO₂-SO₃Li/PVDF-HFP | O₂ exhibits bright with metallic luster (Fig. R5a), and its SEM morphology shows smooth without obvious dendrite growth (Fig. R5d-e). While the anode of Li | SiO₂/PVDF-HFP | O₂ turns black (Fig. R5b) and appears rough surface (Fig. R5f-g). Furthermore, as displayed in Fig. R5c, the lithium anode of Li | PVDF-HFP | O₂ even turns pulverized. These results further confirm that the NNFQSE SiO₂-SO₃Li/PVDF-HFP can not only trap the liquid electrolyte inside to prevent the

side reaction with cathode and reduce the residue of solid-state products, but also suppress lithium dendrite growth.

Fig. R4. Surface morphologies of cathode after cycling. (a) Li | SiO₂-SO₃Li/PVDF-HFP | O₂; (b) Li | SiO₂/PVDF-HFP | O₂; (c) Li | PVDF-HFP | O₂.

Fig. R5 Digital photos and surface morphologies of lithium anode after cycling. (a, d, e) Li | SiO₂-SO₃Li/PVDF-HFP | O₂ ; (b, f, g) Li | SiO₂/PVDF-HFP | O₂; (c) Li | PVDF-HFP | O₂.

Updates to the revised manuscript and supplementary information: We have added these results in **supplementary Figs. 20, 24 and supplementary Note 2,** details in the **revised manuscript on Page 20 (Line 360-363) and Page 21 (Line 396-400), highlighted in yellow.**

Reviewer #2 (Remarks to the Author):

In this work, the authors reported that using SiO₂-SO₃Li/PVDF-HFP as non-Newtonian fluid quasi-solid electrolyte in Li-O₂ battery can effectively eliminate dendrite issues during cycling measurements along with good liquid electrolyte retention and Li⁺ conductivity. As the results, the battery can perform excellent cycling stability of 5000 hours at 100 mA g⁻¹.

Response: We thank the reviewer for the positive comments on the impact of our work. We have revised the manuscript and answered questions following your comments.

Some issues are addressed as follows:

1. It is really hard to see that the S are uniformly distributed on SiO₂-SO₃Li particles from EDS-mapping result in Figure S3. Please check the result and give more clear demonstration.

Response: Thanks for the comments. We have carried out more clear demonstration by elemental mapping of TEM. The images in Fig. R6 can further verify the S are uniformly distributed on SiO₂-SO₃Li particles.

Fig. R6 HADDF-STEM images and the corresponding elemental mapping of SiO₂-SO₃Li particles.

Updates to the revised supplementary information: We have added the results in **Supplementary Fig. 5.**

2.The FTIR results should give more clarifications between the SiO₂, SiO₂-SO₃H and SiO₂-SO₃Li to prove the varied surface chemistry. It is claimed by author that the peak of Si-OH in 2347 cm⁻¹ disappears in SiO₂-SO₃Li compared to SiO₂ due to sulfonated treatment, but why there is a peak of Si-OH at 1632 cm⁻¹ in both SiO₂ and SiO₂-SO₃Li samples in Figure 2d and FigureS5? What the difference between these vibration features?

Response: Thanks for the valuable suggestion. As shown in Fig. R7, we have retested the FTIR spectra of SiO₂. The peak in 2347 cm⁻¹ also disappears, which should be corresponded to CO₂. Moreover, the peak at 1632 cm⁻¹ may corresponded to the bending vibration of -OH in H₂O. In addition, the region of 1000-1300 cm⁻¹ corresponding to Si-O-Si stretching adsorption turns widen both in SiO₂-SO₃H and SiO₂-SO₃Li, and the characteristic peak of -SO₃⁻ stretching at 1190 cm⁻¹ indicates the sulfonated treatment.

Fig. R7 FTIR spectra of porous SiO₂, SiO₂-SO₃H and SiO₂-SO₃Li nanoparticles.

Updates to the revised manuscript: We have added these details in the revised manuscript on Page 7 (Line 125-131, highlighted in yellow) and corrected on Supplementary Fig. 6.

3. Please explain why the core level XPS spectra of S 2p seems very poor signal but showed much stronger intensity in survey spectra in Figure 2e.

Response: Thanks for pointing out the mistake. We have carried out another XPS analysis of SiO₂-SO₃H to ensure the the position of sulfur (Fig. R8b). After adding the fillers into polymer, there was still a peak at the position, but the signal is weakened. The result was corrected the S 2p signal in XPS survey spectra in Fig. R8 and revised in Fig. 2e.

Fig. R8 XPS spectrum of $\text{SiO}_2\text{-SO}_3\text{Li/PVDF-HFP}$ and $\text{SiO}_2\text{-SO}_3\text{H}$.

4. The porosity of control-group samples should be provided to prove the superior microstructure properties of skeleton matrix materials.

Response: Thanks for the valuable suggestion. Mercury intrusion porosimetry (MIP) measurements were carried out to characterize porous features. As shown in Table R1, PVDF-HFP is observed with the highest porosity ratio. After adding SiO_2 nanofillers into PVDF-HFP, the porosity ratio is decreased from 74.6% to 58.4%, and the volume density is increased from 0.34 g mL^{-1} to 0.59 g mL^{-1} , which further proves that the pores of PVDF-HFP are blocked by unmodified SiO_2 , corresponding with the SEM observations (Supplementary Fig. 2c-d). For $\text{SiO}_2\text{-SO}_3\text{Li/PVDF-HFP}$, the porosity ratio, volume density, apparent (skeleton) density and total pore volume all fall in between PVDF-HFP and $\text{SiO}_2\text{/PVDF-HFP}$, revealing that the sulfonate fillers $\text{SiO}_2\text{-SO}_3\text{Li}$ can make the skeleton matrix maintain certain porosity without being blocked. In addition, it is noticed that the total pore area increases from $7.56 \text{ m}^2 \text{ g}^{-1}$, $16.06 \text{ m}^2 \text{ g}^{-1}$ to $18.23 \text{ m}^2 \text{ g}^{-1}$, respectively, owing to the wrinkled radial structure and beautiful folded spherical morphology of SiO_2 nanofillers (Supplementary Fig. 3) and the surface channels of $\text{SiO}_2\text{-SO}_3\text{Li}$ have become deeper and richer after being modified (Fig. 2b-c) (in accordance with Fig. R12).

Table R1 Porous features of PVDF-HFP, SiO₂/PVDF-HFP and SiO₂-SO₃Li/PVDF-HFP.

Electrolyte	Porosity ratio (%)	Volume density (g ml ⁻¹)	Apparent (skeleton) density (g ml ⁻¹)	Total pore volume (mL g ⁻¹)	Total pore area (m ² g ⁻¹)
PVDF-HFP	74.6	0.34	1.34	2.19	7.57
SiO ₂ /PVDF-HFP	58.4	0.59	1.43	0.98	16.06
SiO ₂ -SO ₃ Li /PVDF-HFP	62.9	0.48	1.28	1.32	18.27

Updates to the revised manuscript: We have added these details in the revised manuscript on Page 6 (Line 100-116, highlighted in yellow) and supplementary Table1.

5. Please explain why the stable electrochemical window of SiO₂-SO₃Li/PVDF-HFP is much higher than the others? Besides, LSV with same scan rate to low potential area should also provide to examine its reduction behavior.

Response: Thanks for the valuable comments. The samples tested by LSV are the composite quasi-solid electrolyte which have absorbed liquid electrolyte (LE) inside. The improvement of the stable electrochemical window of SiO₂-SO₃Li/PVDF-HFP may be attributed to the intermolecular interaction between the composite polymer and LE. (*Adv. Mater.* 2019, 31, 1902029; *Adv. Mater.* 2022, 2110423) ⁷Li-NMR spectrum was performed to prove the interaction. As shown in Fig. R9, SiO₂-SO₃Li/PVDF-HFP shows the minimum chemical shift. The variation trend is consistent with LSV in Fig. 3i, which indicates the interactions have changed the chemical environment of composite polymer and LE and then tuned the electrochemical window. The chemical down shift means the shielding effect is enhanced and the electron cloud density around lithium nucleus is increased. The

lithium nucleus in SiO₂-SO₃Li/PVDF-HFP has the largest electron cloud density. That means the environment (such as anions, solvent molecules, et al.) around the lithium nucleus affords extra electrons to Li⁺ and the cloud density of the environment except Li⁺ is reduced, which enhances the oxidation potential. This may lead to the highest oxidation potential of SiO₂-SO₃Li/PVDF-HFP. In addition, the interactions make more liquid electrolyte anchor inside the skeleton of SiO₂-SO₃Li/PVDF-HFP and reduce the contact with electrode. So, the oxidation decomposition and parasitic reaction of liquid electrolyte are also inhibited to some extent.

To examine its reduction behavior, another LSV test of was conducted from 3.5 V to 0 V with the same scan rate 1 mV s⁻¹ (Fig. R10). PVDF-HFP exhibits three peaks at ~1.52 V, ~1.17 V and ~0.77 V, which are indexed as the LiTFSI and TEGDME decomposition peaks of liquid electrolyte, respectively, as reported before. (*Angew. Chem. Int. Ed.* 2022, e202115909) Therein, the 1.52 V peak accounts for the initial dissociation of C-F or S-F bonds, and the 1.17 V peak is the signal of further decomposition of the defluorinated TFSI anions. It worth noting that the cathodic peaks of SiO₂/PVDF-HFP turns lower to 1.49 V, 1.09 V and 0.77 V. And SiO₂-SO₃Li/PVDF-HFP shows the lowest reduction potential at 1.38 V, 1.02 V and 0.65V, which indicates that the Li salt and solvent are absorbed inside SiO₂-SO₃Li/PVDF-HFP and react with the components. As a consequence, SiO₂-SO₃Li/PVDF-HFP may delay the decomposition of liquid electrolyte and ensure the battery safety furtherly.

Fig. R9. NMR analysis of the reaction between composite polymer and liquid electrolyte (LE).

Fig. R10. LSV profiles with same scan rate 1 mV s^{-1} to low potential area, and orange arrows indicate the cathodic peaks where salts decompose.

6. The sample of $\text{SiO}_2/\text{PVDF-HFP}$ showed relatively lower voltage hysteresis compared to $\text{SiO}_2\text{-SO}_3\text{Li}/\text{PVDF-HFP}$ in Figure 4b, c. If reaction kinetics of Li^+ in $\text{SiO}_2\text{-SO}_3\text{Li}/\text{PVDF-HFP}$ is higher than other, why this phenomenon occurred?

Response: Thanks for the comment. Owing to the thickness of composite electrolyte wasn't normalized after swelling, the $\text{SiO}_2/\text{PVDF-HFP}$ in Fig. 4b seems behave lower voltage hysteresis. After retested and normalized, the $\text{SiO}_2/\text{PVDF-HFP}$ shows recover higer voltage hysteresis compared to $\text{SiO}_2\text{-SO}_3\text{Li}/\text{PVDF-HFP}$ (Fig. R11). The results have been revised on Manuscript (Fig. 4b and Fig. 4c).

Fig. R11. Cycling performance of Li-O_2 cells at room temperature.

Reviewer #3 (Remarks to the Author):

This work reported a composite quasi-solid electrolyte SiO₂-SO₃Li/PVDF-HFP with non-Newtonian fluid properties for Li-O₂ batteries (LOBs). Intensive mechanical and electrochemical characterizations were conducted to investigate the properties of the designed electrolyte. The assembled LOBs exhibited an ultra-long cycling life. But the current manuscript lacks analyses and discussions regarding battery chemistry and anode/electrolyte/cathode interfaces. The promotion effect was oversimplified by component interactions. There are also some problems to be clarified before the final acceptance.

Response: We thank the reviewer for the positive comments on the impact of our work. We have revised the manuscript and answered questions following your comments.

1. The author calculated the contact angle of SiO₂-SO₃Li/PVDF-HFP with a different tangent center from control samples (Fig. S7). A wrong conclusion about wettability was obtained. From the direct view, the contact angles of the three samples are similar. The enhanced wettability of SiO₂-SO₃Li/PVDF-HFP is doubting.

Response: Thanks for pointing out the mistake. I was careless to mix up the images of contact angle when typesetted. We have switched back the original data and revised in **Supplementary Fig. 8**.

2. The order of Figures in supporting information is inconsistent with the manuscript.

Response: Thanks for pointing out and comment. We have corrected the orders.

3. The preparation procedure of the cathode should be illustrated in the experimental section.

Response: Thanks for the suggestion. We have added the related preparation procedure of the cathode in the experimental section (**Manuscript Page 24, Line 458-464**).

4. From presented photos of SiO₂-SO₃Li/PVDF-HFP before and after cycling, the cycled electrode became thicker than the initial, and why?

Response: Thanks for the comment. Before cycling, the SiO₂-SO₃Li/PVDF-HFP was thick as 100 μm (Supplementary Fig. 1a-b). After full swelling, its thickness increased to 112 μm. Coupling with the shearing action during plating/stripping process, SiO₂-SO₃Li/PVDF-HFP may turn softer and change to elastic jelly (Supplementary Fig. 17a), which result in the final 116 μm thick after cycling.

5. From a recent literature review of LOBs (doi.org/10.1039/D2CS00003B), the battery chemistry is significantly affected by the electrolyte. How the SiO₂-SO₃Li/PVDF-HFP electrolyte affects the growth and decomposition of discharge products is yet lacking, as well as the crystal structure and morphology.

Response: Thanks for the valuable comments. In order to better understand how the SiO₂-SO₃Li/PVDF-HFP electrolyte affects the growth and decomposition of discharge products, we investigated the morphology and crystal structure of the cathode after discharge and charge. As shown in Fig. R12, high-magnification SEM images reveal that the discharge products of Li₂O₂ are formed on the cathode (Fig. R12a-b) and completely disappeared after charging (Fig. R12c-d), demonstrating good reversibility during the charge/discharge process. Moreover, X-ray diffraction (XRD) analyses of the pristine, discharged, and charged cathodes were carried out in Fig. R12e. Due to the hollow tubular structure of cathode, the XRD signal is shielded to a certain extent. The discharged cathode exhibits two weak peaks located at 32.9° and 35.8°, which correspond to the (100) and (101) crystal surfaces of Li₂O₂, respectively. (*Adv. Energy Mater.* 2017, 1701203) After charging, the peaks completely disappear and recover to be consistent with the pristine crystal structure. These results perfectly validate the SiO₂-SO₃Li/PVDF-HFP electrolyte exhibits robust chemical stability against products during cycling. (*Chem. Soc. Rev.*, 2022, 51, 8045–8101)

Figure R12. Characterization of the growth and decomposition of discharge products. (a-b) SEM images showing the Li_2O_2 is formed on the cathode after discharge; (c-d) SEM images showing the Li_2O_2 formed on the cathode has vanished after charge. (e) XRD analysis of cathode after discharge and charge.

Updates to the revised manuscript and supplementary information: We have added these results in Supplementary Fig. 23 and Supplementary Note 2, details in the revised manuscript on Page 21 (Line 396-397), highlighted in yellow.

6. Although $\text{SiO}_2\text{-SO}_3\text{Li/PVDF-HFP}$ exhibited a wide electrochemical window, reactive oxygen species generated during the discharging and charging of LOBs could also induce the decomposition of the electrolyte and functional groups, especially during ultra-long operation. Relevant discussions and analyses are recommended in order to obtain a more comprehensive understanding of the stability of the designed electrolyte.

Response: Thanks for the valuable comments. Actually, the $\text{SiO}_2\text{-SO}_3\text{Li/PVDF-HFP}$ electrolyte maintains stable during long-term Li- O_2 cycling, which can be supported by the FTIR, XPS and XRD analysis before and after cycling. It is obvious that there no new vibration peaks are generated in the FTIR spectrum (Fig. 5c) and no new functional groups appear in the XPS spectra (Fig. 5d-f) of $\text{SiO}_2\text{-SO}_3\text{Li/PVDF-HFP}$ electrolyte after 5000 h cycling. In addition, the XRD analysis exhibits the same crystal structure as before without any extra phase (Fig. R13).

Therefore, the designed electrolyte performs well stability during ultra-long operation.

Fig. R13. XRD analysis of SiO₂-SO₃Li/PVDF-HFP after 5000 h Li-O₂ cycling.

Updates to the revised manuscript: We have added these details in the revised manuscript on Page 18 (Line 329-332), highlighted in yellow.

7. As shown in Figure S14, the impedance of the cycled LOB increased significantly. This could be caused by the coverage of undecomposed discharge products or the degradation/evaporation of the electrolyte. The actual reasons should be clarified.

Response: Thanks for the comments. As shown in Fig. R4 and Fig. R5 mentioned above, the cathode of Li | SiO₂-SO₃Li/PVDF-HFP | O₂ exists little residue of solid-state products and some area of its anode turns black after long-term cycling, which may result in the increased impedance.

8. From Figure 2c and Figure S2c, similar morphology was observed on SiO₂-SO₃Li and SiO₂. How did the author define the “surface channels” and conclude the “deeper and richer” surface channels of SiO₂-SO₃Li than SiO₂?

Response: Thanks for the comments. As compared, the TEM image of SiO₂-SO₃Li particle in Fig. 2c shows surface channels with 60.7 nm, which is more than twice deeper and richer than SiO₂ particle in supplementary Fig. 3c (25 nm). We have adjusted the order of Fig. S2c to Supplementary Fig. 3c.

Fig. R14. TEM images of (a) SiO₂-SO₃Li particle in Fig. 2c and (b) SiO₂ particle in supplementary Fig. 3c.

Updates to the revised manuscript: We have added these details in the revised manuscript on Page 6 (Line 113-116), Fig. 2c and Supplementary Fig. 3c, highlighted in yellow.

9. The introduction of the -SO₃Li group significantly changed the morphology of composite polymers, enabling uniform distribution of SiO₂ and providing Li⁺ diffusion channels. But the extra roles of the -SO₃Li groups in promoting battery performance are ambiguous and lack compelling evidence.

Response: Thanks for the comments. In addition to improving the morphology of composite polymers, enabling uniform distribution of SiO₂ and providing Li⁺ diffusion channels, our experimental results

also verified that the $-\text{SO}_3\text{Li}$ groups contribute to higher electrolyte uptake and retention ratio (see Fig. 3a-b), which may ensure the liquid electrolyte would not leak and consume out during long-term cycling in the semi-open cell system, and thus increase the battery lifespan. Besides, Fig. 3g directly proves the significant enhancement of Li^+ conductivity, indicating higher reaction kinetics of Li^+ in $\text{SiO}_2\text{-SO}_3\text{Li/PVDF-HFP}$ and lower polarization voltage in batteries. These statements have been verified through previous reports, which demonstrated that the sulfonate groups $-\text{SO}_3\text{Li}$ with electronegativity can simultaneously facilitate access by the electrolyte to promote ion pair dissociation and increase the mobility of Li^+ . (*Adv. Mater.* 2021, 33, 2105178) $-\text{SO}_3\text{Li}$ sites also provide the lithium ion hopping pathways to improve Li^+ conductivity. (*Nano Lett.* 2021, 21, 2997–3006; *J. Am. Chem. Soc.* 2019, 141, 5880–5885) Furthermore, with $-\text{SO}_3\text{Li}$ groups, the designed electrolyte behaves the non-Newtonian fluid characteristics of shear thinning and shear thickening at the same time. This characteristic has more significant gain in cycle life, giving the electrolyte unparalleled long cycle performance.

REVIEWERS' COMMENTS

Reviewer #1 (Remarks to the Author):

The questions are more clearly explained and should be published in this version.

Reviewer #2 (Remarks to the Author):

In this work, the authors reported that using SiO₂-SO₃Li/PVDF-HFP as non-Newtonian fluid quasi-solid electrolyte in Li-O₂ battery can effectively eliminate dendrite issues during cycling measurements along with good liquid electrolyte retention and Li⁺ conductivity. As the results, the battery can perform excellent cycling stability of 5000 hours at 100 mA g⁻¹. The revised version of manuscript can be acceptable for publication. A minor issue is addressed as follows:

Comment 1: Author should carefully notice that most electrolyte used in battery is non-Newtonian electrolyte. More precise wording should be used to distinguish whether it is under shear-thickening region or shear-thinning.

Reviewer #3 (Remarks to the Author):

I happy with the authors' response. No further questions from my side.

Point-by-point Response to the Reviewers

REVIEWERS' COMMENTS

Reviewer #1 (Remarks to the Author):

The questions are more clearly explained and should be published in this version.

Response: Thanks for the positive comments.

Reviewer #2 (Remarks to the Author):

In this work, the authors reported that using $\text{SiO}_2\text{-SO}_3\text{Li/PVDF-HFP}$ as non-Newtonian fluid quasi-solid electrolyte in Li-O₂ battery can effectively eliminate dendrite issues during cycling measurements along with good liquid electrolyte retention and Li⁺ conductivity. As the results, the battery can perform excellent cycling stability of 5000 hours at 100 mA g⁻¹. The revised version of manuscript can be acceptable for publication. A minor issue is addressed as follows:

Comment 1: Author should carefully notice that most electrolyte used in battery is non-Newtonian electrolyte. More precise wording should be used to distinguish whether it is under shear-thickening region or shear-thinning.

Response: Thanks for the positive comments and the suggestion. The following descriptive words have been added in manuscript (Page 8, Line 151-154, highlighted in yellow):

“It is usually agreed that when $G' > G''$, the materials will mechanically stiffen and exhibit shear-thickening property. On the contrary, when $G'' > G'$, the materials will turn soft and exhibit shear-thinning. As shown in Fig. 2f, at relatively low strain rate, ...”.

Reviewer #3 (Remarks to the Author):

I happy with the authors' response. No further questions from my side.

Response: Thanks for the positive comments.